# RefinePose: Towards More Refined Human Pose Estimation

Hao Dong, Guodong Wang *, Chenglizhao Chen and Xinyue Zhang

College of Computer Science and Technology, Qingdao University, Qingdao 266000, China
* Correspondence: doctorwgd@gmail.com

**Abstract:** Human pose estimation is a very important research topic in computer vision and attracts more and more researchers. Recently, ViTPose based on heatmap representation refreshed the state of the art for pose estimation methods. However, we find that ViTPose still has room for improvement in our experiments. On the one hand, the PatchEmbedding module of ViTPose uses a convolutional layer with a stride of $14 \times 14$ to downsample the input image, resulting in the loss of a significant amount of feature information. On the other hand, the two decoding methods (Classical Decoder and Simple Decoder) used by ViTPose are not refined enough: transpose convolution in the Classical Decoder produces the inherent chessboard effect; the upsampling factor in the Simple Decoder is too large, resulting in the blurry heatmap. To this end, we propose a novel pose estimation method based on ViTPose, termed RefinePose. In RefinePose, we design the GradualEmbedding module and Fusion Decoder, respectively, to solve the above problems. More specifically, the GradualEmbedding module only downsamples the image to $1/2$ of the original size in each downsampling stage, and it reduces the input image to a fixed size ($16 \times 112$ in ViTPose) through multiple downsampling stages. At the same time, we fuse the outputs of max pooling layers and convolutional layers in each downsampling stage, which retains more meaningful feature information. In the decoding stage, the Fusion Decoder designed by us combines bilinear interpolation with max unpooling layers, and gradually upsamples the feature maps to restore the predicted heatmap. In addition, we also design the FeatureAggregation module to aggregate features after sampling (upsampling and downsampling). We validate the RefinePose on the COCO dataset. The experiments show that RefinePose has achieved better performance than ViTPose.

**Keywords:** human pose estimation; ViTPose; vision transformer; heatmap; deep learning

## 1. Introduction

Two-dimensional human pose estimation (or human keypoint detection) is the process of locating human joints in a given image. With the rapid development of artificial intelligence and increasingly downstream industrial projects, human pose estimation has also ushered in a golden age of vigorous development. Because of the wide range of application scenarios (such as human–computer interaction, automatic driving, action special effects, etc.), human pose estimation is a vital research topic in the field of computer vision.

Recently, ViTPose [1], which is very prevalent in the field of human pose estimation, has aroused our research interest. It is a top-down approach using heatmap representation and achieves state-of-the-art (SOTA) performance. ViTPose is developed based on ViT [2], so its design is not complicated. The authors believe that the potential of plain ViT is giant, despite requiring enormous data support. Experiments show that the highest AP value of ViTPose using MAE reaches 79.1 on the MS COCO dataset [3].

Although ViTPose has impressive performance, we also find some shortcomings during our experiments. Firstly, the PatchEmbedding module in ViTPose directly compresses the height and width of input images to $1/16$ of the original, which loses a lot of feature information. Taking the input image with $256 \times 192$ as an example, the PatchEmbedding module scales the input image from $256 \times 192$ to $16 \times 12$. Such a high compression ratio will lose a lot of feature information, which directly limits the learning of the subsequent

transformer modules. Secondly, the two decoding methods adopted by ViTPose in the decoding stage (converting the feature map into a predicted heatmap) have shortcomings: (1) Using bilinear interpolation with a 4x upsampling factor results in an inaccurate predicted heatmap. Because the upsampling factor is too large, the enlarged heatmap is blurry [4,5]. (2) Using transposed convolution with stride 2 × 2 to upsample twice will cause chessboard effect [6]. (3) The stacking number of the transformers in ViTPose is very large (for example, the stacking number in huge-version ViTPose reaches 32). If the training set is not large enough, the problem of gradient disappearance is easy to occur.

To solve the above problems, we design a new human pose estimation framework based on ViTPose [1], termed RefinePose. First, we innovatively design the GradualEmbedding module to replace the original PatchEmbedding. The GradualEmbedding module only scales down the image to 1/2 of the original size (instead of 1/16 of the PathEmbeding) at each step, and gradually downsamples to the specified size through multiple steps. This progressive downsampling strategy can decrease the feature loss in the downsampling process. At the same time, we combine the MaxPool layer with stride 2 × 2 and the convolutional layer with stride 2 × 2 to achieve downsampling. The neural network can not only learn feature information but also strengthen significant features to retain more meaningful information during the downsampling process. Second, the Fusion Decoder is proposed to decode more refined heatmaps. Under the guidance of the indices returned by MaxPool in GradualEmbedding, the Fusion Decoder combines bilinear interpolation with the MaxUnpool to realize more refined upsampling. Meanwhile, we introduce depthwise separable convolution [7] to aggregate features after upsampling. Third, we add a shortcut strategy to the stacking module of transformers, which can alleviate the problem of insufficient feature extraction on small datasets.

To sum up, the contributions of this paper can be summarized as follows:

1.  We design a novel GradualEmbedding module, which uses a progressive downsampling strategy to scale down input images. Additionally, it combines convolutional layers and MaxPool layers to retain more meaningful feature information during the downsampling process.
2.  We propose a more refined decoding method, termed Fusion Decoder. It combines bilinear interpolation and MaxUnpool together to decode more refined heatmaps.
3.  We add shortcuts to the stacking module of transformers to alleviate insufficient feature extraction on small datasets.

## 2. Vision Transformer for Human Pose Estimation

With the rapid development of deep learning, human pose estimation has also made significant progress. Since the Transformer [8] was proposed in 2017, it has attracted the attention of numerous computer vision researchers due to its excellent performance. At present, a lot of excellent works [2,9–15] based on the Transformer [8] or its variants [16–20] have emerged. There are also many excellent works [21–26] based on Transformer in the field of human posture estimation. PRTR [21] utilizes the encoder–decoder structure of transformers to perform a regression-based person and keypoint detection. Compared with the existing methods, it requires less heuristic design and is a universal detection architecture. The authors use a cascade strategy to gradually refine the coordinates of keypoints and obtain excellent experimental results. PoseTrans [27], TokenPose [28], and TransPose [22] all use transformers to build a pose estimation framework on the feature maps extracted by CNNs. TokenPose can effectively predict the locations of occluded keypoints and model the relationship among different keypoints by introducing token design. TransPose proves that the keypoint location methods based on transformers conform to the interpretability of activation maximization. HRFormer [26] proposes a transformer-based network that can learn high-resolution features. HRFormer is improved from HRNet [29]. The author adopts a multi-resolution parallel design and local-window attention mechanism to perform attention computations on non-overlapping windows to improve the calculation efficiency. In addition, the authors also introduce convolution

operations in the FFN module to exchange information among different image windows. ViTPose [1], as the outstanding human pose estimation method this year, has achieved the highest accuracy on the MS COCO dataset. The author believes that plain ViT [2] has great potential. ViTPose does not use CNNs to extract features, nor does it have a complex network design. As long as it has enough training data, it can achieve good performance. The experimental results also confirm the excellent performance of ViTPose, which achieves SOTA accuracy on the MS COCO dataset.

## 3. Method

Xu et al. [1] designed ViTPose to explore the potential of plain ViT in human pose estimation tasks. Although ViTPose has achieved SOTA performance, we think that there is still room for improvement in its architectural design. For example, the PatchEmbedding module greatly compresses the image size, which leads to the loss of feature information, and the chessboard effect caused by deconvolution in the decoding stage. To solve the above problems, we design a new human pose estimation framework, termed RefinePose. RefinePose is an improved version based on ViTPose [1], and the details of its architecture are shown in Figure 1. Our proposed GradualEmbedding module and Fusion Decoder are core parts of the RefinePose. More specifically, the GradualEmbedding module can alleviate the problem of feature information loss in the downsampling process; the newly designed Fusion Decoder can avoid the checkerboard effect. Next, we will introduce each part of RefinePose in more detail.

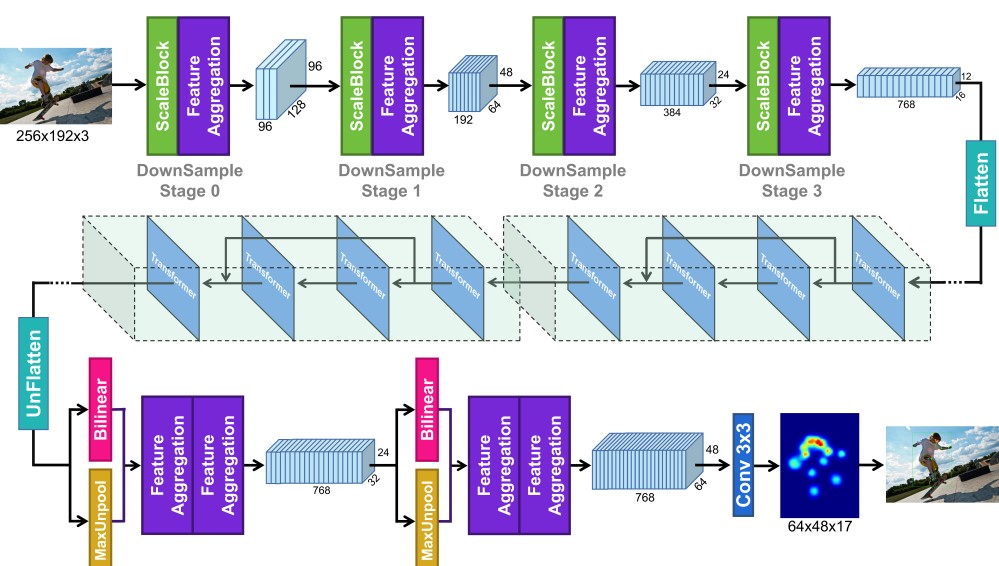

**Figure 1.** The architectural overview of the RefinePose.

### 3.1. The Proposed GradualEmbedding

The PatchEmbedding module in ViTPose mainly converts RGB images ($H \times W \times 3$) into two-dimensional tokens ($N \times C$), and then the model sends these tokens to the subsequent transformers for learning features. This process mainly involves two operations: downsampling the feature maps and flattening the feature maps into sequential tokens. Take the image with $256 \times 192 \times 3$ as an example: in PatchEmbedding, ViTPose first uses a convolutional layer with $16 \times 16 \times 3$ kernels and $16 \times 16$ stride to reduce the input image to $1/16$ of the original, obtaining the feature map with $16 \times 12 \times 768$; then the model flattens the feature map ($16 \times 12 \times 768$) into sequential tokens ($192 \times 768$). This process can be expressed by the following equations:

$$y = Conv(x) \tag{1}$$

$$z = Flatten(y) \tag{2}$$

where $x$ represents the input images and $Conv(*)$ and $Flatten(*)$ represent convolution operation and flatten operation, respectively. Meanwhile, $x \in \mathbb{R}^{H \times W \times 3}$, $y \in \mathbb{R}^{\frac{H}{16} \times \frac{W}{16} \times C}$, $z \in \mathbb{R}^{N \times C}$, $N = H \times W$, where $H$, $W$, and $C$ represent the height, width, and channel of images, respectively.

From Equation (1), we can see that the convolution operation in PatchEmbedding directly downsamples the $256 \times 192$ images to $16 \times 12$. Although the channel of feature maps has increased from 3 to 768, the convolutional layer with $16 \times 16 \times 3$ kernel size converts the input image of $256 \times 192 \times 3$ into a feature map of $16 \times 12 \times 1$. This convolution operation with a large stride will result in a large loss of feature information. In order to alleviate the loss of numerous feature information, we design GradualEmbedding to replace PatchEmbedding in ViTPose. The details of the GradualEmbedding module are shown in Figure 2, and the detailed comparison between GradualEmbedding and PatchEmbedding is shown in Figure 3.

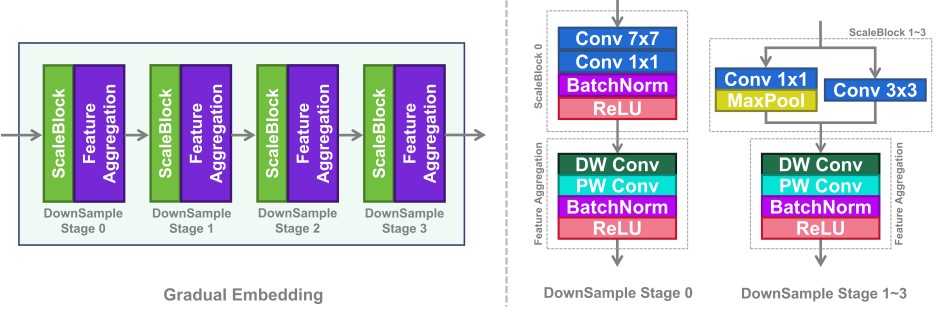

**Figure 2.** The details of the GradualEmbedding module.

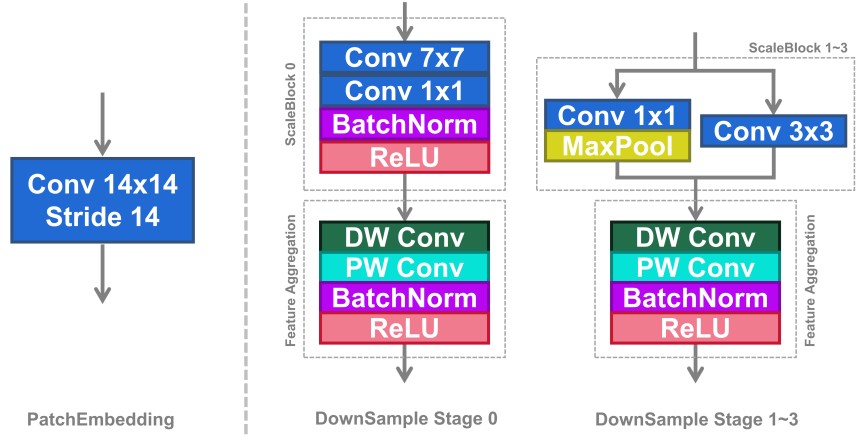

**Figure 3.** Comparison between the PatchEmbedding and our GradualEmbedding.

As can be seen from Figure 2, the GradualEmbedding module designed by us is a progressive downsampling structure. The GradualEmbedding module contains four downsampling blocks, and each downsampling block includes two parts: the ScaleBlock and the FeatureAggregation. Among them, the ScaleBlock is responsible for the specific downsampling task, reducing the size of feature maps to 1/2 of the original size; FeatureAggregation is responsible for aggregating features on the reduced feature maps. To reduce the number of parameters, we use depthwise separable convolution [7] to replace the traditional convolutional layer in FeatureAggregation. The GradualEmbedding module part can be represented by the following equations:

$$x_1 = DownSample_1(x_{input}) \tag{3}$$

$$x_{i+1} = DownSample_i(x_i), i \in [1, 2, 3] \tag{4}$$

$$y = FinalConv(x_4) \tag{5}$$

$$z = Flatten(y) \tag{6}$$

where $x_i$, $y$, and $z$ represent the feature maps. $FinalConv(*)$ represents the "Conv-Batch-Norm" block, and $Flatten(*)$ represents flatten operation. $DownSample_i(*)$ is represented by the following equations:

$$ScaleBlock(x) = Conv_{3x3}(x) + MaxPool(Conv_{1x1}(x)) \tag{7}$$

$$FeatureAggregation(x) = ReLU(BatchNorm((DSC(x)))) \tag{8}$$

$$DownSample_1(x) = FeatureAggregation(Conv_{1x1}(Conv_{7x7}(x))) \tag{9}$$

$$DownSample_i(x) = FeatureAggregation(ScaleBlock(x)), i \in [2,3,4] \tag{10}$$

where $Conv_{k \times k}(*)$ and $MaxPool(*)$ represent the convolutional layer with $k \times k$ kernel size and max pooling layer, respectively. $DSC(*)$ represents the depthwise separable convolution [7].

As can be seen from Equation (7), the ScaleBlock designed by us is obtained by adding the results of $3 \times 3$ convolutional layer and MaxPool layer. It is worth noting that the strides of the $3 \times 3$ convolutional layer and MaxPool layer are both set to 2 to achieve the purpose of downsampling the feature maps. The former extracts feature information while reducing the size of feature maps; the latter extracts the maximum response points of feature maps while reducing the size of feature maps. The two parts are added to obtain the final downsampled feature maps. If the size of the feature map $x_i$ is $H \times W \times C$, then the size of the feature map $x_{i+1}$ after downsampling is $\frac{H}{2} \times \frac{W}{2} \times 2C$. This design allows the neural network to reversely strengthen the significant features while learning the feature information, effectively alleviating the problem of feature information loss in the downsampling process and retaining more meaningful feature pixels. The FeatureAggregation module is mainly composed of depthwise separable convolution [7]. ReLU and BatchNorm are added to avoid overfitting. The FeatureAggregation module does not modify the size of feature maps, it just aggregates features after downsampling.

It is worth noting that we do not use the ScaleBlock in the first downsampling block, but use a convolutional layer with $7 \times 7$ kernel size for downsampling. This design is borrowed from ResNet [30] and DenseNet [31]. The subsequent three downsampling blocks are composed of the ScaleBlock and FeatureAggregation. This progressive downsampling strategy of GradualEmbedding effectively alleviates the loss of feature information during downsampling. The ablation experiments in Section 4.4.1 forcefully verify the excellent performance of the GradualEmbedding module.

### 3.2. The Proposed Fusion Decoder

The task of the decoding stage is converting the feature maps into heatmaps with the designated size (the size of heatmaps on the COCO dataset [3] is generally $64 \times 48 \times 17$), which is also the last step of keypoint detection task. In the decoding stage, ViTPose adopts two methods: the Classic Decoder and Simple Decoder. In the method of the Simple Decoder, the author first uses bilinear interpolation to directly upsample 4 times ($16 \times 12 \rightarrow 64 \times 48$), and then uses the convolutional layer with $3 \times 3$ kernel size to change the channel number of feature maps; In the method of the Classic Decoder, the author first uses two transposed convolutional layers with $2 \times 2$ stride to upsample twice and then uses a convolutional layer with $1 \times 1$ kernel size to change the channel number of feature maps. We think that the above two decoding methods have drawbacks: the upsampling stride in the Simple Decoder is too large, which leads to inaccurate prediction of heatmaps; the transposed convolution in the Classic Decoder will cause a checkerboard effect [4,5]. Odena et al. point out in [6] that the deconvolution operation makes the generated image have a chessboard effect, and the combination of interpolation and convolution can eliminate this chessboard effect. Therefore, inspired by Odena et al. [6], we design a new decoding method termed the Fusion Decoder.

As shown in Figure 4, the Fusion Decoder does not use transposed convolution to achieve upsampling. Instead, it uses a combination of bilinear interpolation and max unpooling to achieve upsampling. This idea is consistent with the design of GradualEmbedding. More specifically, the Fusion Decoder first uses bilinear interpolation to upsample the feature map, then uses the MaxUnpool layer to upsample the feature map again, and adds the results of the two upsampling to obtain the final upsampled feature map. The depthwise separable convolutions are used to aggregate features after upsampling. The above operations are repeated twice so that the feature map of $16 \times 12 \times 768$ is upsampled to the feature map of $64 \times 48 \times 768$. Finally, a convolutional layer with $3 \times 3$ kernel size is used to convert the feature map into a predicted heatmap of $64 \times 48 \times 17$. The Fusion Decoder can be represented by the following equations:

$$UpSample_i(x) = FA(FA(MaxUnpool(x) + Bilinear(x))) \tag{11}$$

$$x_{i+1} = UpSample_i(x_i), i \in [1, 2] \tag{12}$$

$$x_{output} = Conv_{3\times 3}(x_3) \tag{13}$$

where $MaxUnpool(*)$ and $Bilinear(*)$ represent the max unpooling operation and bilinear interpolation, respectively. $FA(*)$ represents the FeatureAggregation module, and $Conv_{3\times 3}(*)$ represents the convolutional layer with $k \times k$ kernel size.

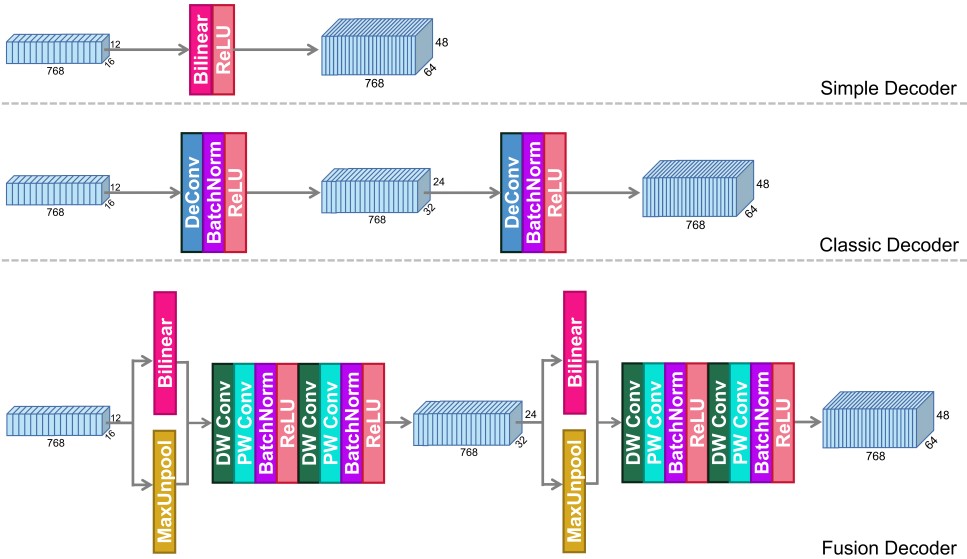

**Figure 4.** Comparison among different decoders including the Fusion Decoder (our), Classic Decoder (ViTPose), and Simple Decoder (ViTPose).

### 3.3. Auxiliary Optimization

Our third improvement on ViTPose is adding shortcuts to the stacking module of transformers to provide sufficient feature information in deep network learning, as shown in Figure 5. The ViTPose stacks 12 transformer modules in the base version and stacks 32 transformer modules in the huge version. We cannot adequately train such a huge network if the dataset is not large enough, and the problem of gradient disappearance is likely to occur. The ResNet model [30] proposes a residual structure, allowing the output of the model to skip some middle layers and directly participate in the training of later layers. This residual method can avoid gradient disappearance and speed up network training. Inspired by ResNet [30], we divide each four transformer modules into a group and add shortcuts to each group. This design can supplement feature information for the deep network. The detailed comparison is shown in Figure 5. We verify the effectiveness of this idea in the ablation experiments in Section 4.4.3.

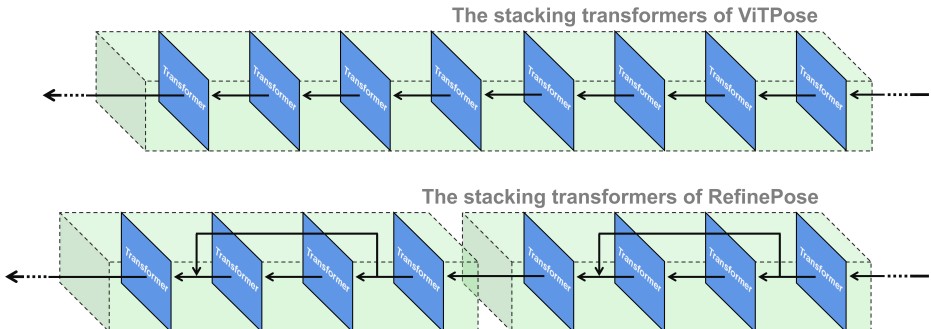

**Figure 5.** Comparison of the stacking transformers between RefinePose and ViTPose.

## 4. Experiments

### 4.1. Datasets

To better demonstrate the performance of RefinePose, we chose the COCO dataset [3] and the MPII dataset [32] for training and testing.

The COCO dataset is a large and rich dataset for object detection and segmentation, which is the mainstream dataset for studying human pose estimation. The 2017 version of the COCO dataset contains a total of 164,000 images, including 118,000 training images, 41,000 test images, and 5000 validation images. In the keypoint detection task, the COCO dataset annotates each person instance with 17 keypoints. We train on the COCO train2017 dataset, test on the test2017 dataset, and evaluate on val2017.

The MPII dataset is a standard 2D human pose dataset, which is based on thousands of YouTube videos. It contains about 24,984 human images and has 16 joint points for each instance. The MPII dataset evenly covers a variety of common and rare human poses, as well as people interacting with various objects and environments. The division method of the training set and validation set is consistent with the HRNet [29].

### 4.2. Implementation Details

The RefinePose is implemented by PyTorch 1.7.0 and runs on an NVIDIA GeForce RTX 3090 GPU. We use the AdamW optimizer and set the learning rate to $5 \times 10^{-3}$. Select the best model from a total of 310 epochs. After cropping a single-person image according to the bounding box, we scale it to a fixed size of $256 \times 192$. In addition, we also perform augmentation operations such as rotation, scaling, and flipping on the dataset. We also use the UDP [33] for pose-processing. The main idea of UDP includes two aspects: one is to use the unit length to measure the size of images instead of the number of pixels during data processing. The other is to introduce an encoding and decoding method without statistical error under ideal conditions; please refer to [33] for more details. It should be noted that the above data augmentation methods do not change the number of images. Therefore, after data augmentation, the number of images used for training and validation is still the number of images in the original train-set and Val-set (118,287 for training and 5000 for validation in COCO 2017).

### 4.3. Experimental Results

#### 4.3.1. Experimental Results on the COCO Dataset

We test the performance of the RefinePose on the MS COCO val2017 and test2017 datasets, respectively. The detailed results are shown in Table 1.

On the val2017 dataset, our proposed RefinePose model with ViT-B backbone achieves 75.9 points AP, which exceeds the classical method HRNet (with HRNet-W32 backbone) by 1.5 points AP at the same resolution. ViTPose with ViT-B backbone, TokenPose-L/D24 with HRNet-W48 backbone, and TransPose-H/A6 with HRNet-W48 backbone all obtain 75.8 points AP, which is 0.1 points AP lower than our RefinePose with the ViT-B backbone. At the resolution of $256 \times 192$, RefinePose also exceeds HRFormer (with HRFormer -B backbone) by 0.3 points AP. When we use the larger ViT-L as the backbone network, the

performance of RefinePose is further improved to 78.3 points AP, which has exceeded most pose estimation methods. With the same backbone (ViT-L) network, RefinePose achieves 0.2 points AP improvements based on ViTPose. At the same resolution, RefinePose surpasses the UDP by 1.3 points AP and HRNet by 3.4 points AP, respectively. When we use the ViT-H as a backbone network, the performance of RefinePose reaches the peak, achieving a pose recognition accuracy of 79.3 points AP. Compared with the ViTPose (with ViT-H backbone) method, our RefinePose achieves 0.2 points AP improvements.

**Table 1.** Comparison with SOTA methods on the MS COCO val and test-dev set.

| Method | Backbone | Input Resolution | COCO val | | COCO test-dev | |
| --- | --- | --- | --- | --- | --- | --- |
| | | | AP | AR | AP | AR |
| HigherHRNet [34] | HRNet-W48 | 384 × 288 | 72.1 | - | 70.5 | 74.9 |
| SimpleBaseline [35] | ResNet-152 | 256 × 192 | 73.5 | 79.0 | - | - |
| HRNet [29] | HRNet-W32 | 256 × 192 | 74.4 | 78.9 | - | - |
| HRNet [29] | HRNet-W32 | 384 × 288 | 75.8 | 81.0 | 74.9 | 80.1 |
| HRNet [29] | HRNet-W48 | 256 × 192 | 75.1 | 80.4 | - | - |
| HRNet [29] | HRNet-W48 | 384 × 288 | 76.3 | 81.2 | 75.5 | 80.5 |
| UDP [33] | HRNet-W48 | 384 × 288 | 77.2 | 82.0 | - | - |
| TokenPose-L/D24 [28] | HRNet-W48 | 256 × 192 | 75.8 | 80.9 | 75.1 | 80.2 |
| TransPose-H/A6 [22] | HRNet-W48 | 256 × 192 | 75.8 | 80.8 | 75.0 | - |
| HRFormer [26] | HRFormer-B | 256 × 192 | 75.6 | 80.8 | - | - |
| HRFormer [26] | HRFormer-B | 384 × 288 | 77.2 | 82.0 | 76.2 | 81.2 |
| ViTPose [1] | ViT-B | 256 × 192 | 75.8 | 81.1 | 75.1 | 80.3 |
| ViTPose [1] | ViT-L | 256 × 192 | 78.3 | 83.5 | 77.3 | 82.4 |
| ViTPose [1] | ViT-H | 256 × 192 | 79.1 | 84.1 | 78.1 | 83.1 |
| RefinePose | ViT-B | 256 × 192 | 75.9 | 82.1 | 75.3 | 80.5 |
| RefinePose | ViT-L | 256 × 192 | 78.5 | 83.7 | 77.5 | 82.3 |
| RefinePose | ViT-H | 256 × 192 | 79.3 | 84.0 | 78.4 | 83.3 |

On the test2017 dataset, our method still achieves the best performance. RefinePose with the ViT-B backbone reaches an excellent score of 75.3 points AP, which exceeds the classical method HRNet (with HRNet-W32 backbone and 384x288 resolution) by 0.4 points AP. At the same resolution, RefinePose surpasses the TokenPose-L/D24 (with HRNet-W48 backbone) by 0.2 points AP and TransPose-H/A6 (with HRNet-W48 backbone) by 0.3 points AP, respectively. With the same backbone (ViT-B) network, RefinePose achieves 0.2 points AP improvements based on ViTPose. Using a larger backbone network can also bring better recognition accuracy on the test2017 dataset. RefinePose using ViT-L backbone reaches a recognition accuracy of 77.5 points AP, which exceeds the ViTPose with ViT-L backbone by 0.2 points AP. RefinePose achieves a state-of-the-art accuracy of 78.4 points AP when we use the ViT-H backbone network.

4.3.2. Experimental Results on the MPII Dataset

To better verify the performance of RefinePose, we evaluate it on the MPII dataset with the ground truth bounding boxes. The test results are shown in Table 2. We follow the default settings of the MPII dataset, and we use the PCKh metric to evaluate the performance of RefinePose. The resolution of the input images is 256 × 256.

On the MPII dataset, RefinePose proposed by us with ViT-B backbone achieves 93.6 average PCKh, which exceeds the classical method HRNet (with HRNet-W32 backbone) by 3.6 PCKh. With the same backbone (ViT-B) network, ViTPose achieves 93.4 average PCKh, which is lower than RefinePose by 0.2 PCKh. Meanwhile, the accuracy of RefinePose also exceeds the CFA and ASDA by 3.5 PCKh and 2.2 PCKh, respectively. In addition, the prediction accuracy of the "Head" is the highest, reaching 97.8 PCKh; the prediction accuracy of "Ankle" is the lowest, reaching 89.4 PCKh. When we use the larger ViT-L as the backbone network, the performance of RefinePose is further improved to 94.1 PCKh.

RefinePose achieves 0.2 PCKh improvements based on ViTPose. RefinePose achieves a state-of-the-art accuracy of 94.2 PCKh when we use the ViT-H backbone network.

**Table 2.** Comparison with SOTA methods on the MPII dataset (PCKh@0.5).

| Method | Backbone | Head | Shoulder | Elbow | Wrist | Hip | Knee | Ankle | Mean |
|---|---|---|---|---|---|---|---|---|---|
| CPM [36] | CPM | 96.2 | 95.0 | 87.5 | 82.2 | 87.6 | 82.7 | 78.4 | 87.7 |
| SimpleBaseline [35] | ResNet-152 | 86.9 | 95.4 | 89.4 | 84.0 | 88.0 | 84.6 | 82.1 | 89.0 |
| HRNet [29] | HRNet-W32 | 96.9 | 85.9 | 90.5 | 85.9 | 89.1 | 86.1 | 82.5 | 90.0 |
| HRNet [29] | HRNet-W48 | 97.1 | 95.8 | 90.7 | 85.6 | 89.0 | 86.8 | 82.1 | 90.1 |
| CFA [37] | ResNet-101 | 95.9 | 95.4 | 91.0 | 86.9 | 89.8 | 87.6 | 83.9 | 90.1 |
| ASDA [38] | HRNet-W48 | 97.3 | 96.5 | 91.7 | 87.9 | 90.8 | 88.2 | 84.2 | 91.4 |
| TransPose-H/A6 [22] | RNet-W48 | - | - | - | - | - | - | - | 92.3 |
| ViTPose [1] | ViT-B | 97.6 | 97.4 | 93.7 | 90.1 | 92.4 | 91.9 | 88.3 | 93.4 |
| ViTPose [1] | ViT-L | 97.7 | 97.4 | 94.0 | 91.5 | 93.1 | 92.2 | 89.7 | 93.9 |
| ViTPose [1] | ViT-H | 97.7 | 97.6 | 94.3 | 91.2 | 93.3 | 92.5 | 90.1 | 94.1 |
| RefinePose | ViT-B | 97.8 | 97.5 | 93.9 | 90.3 | 92.4 | 92.0 | 89.4 | 93.6 |
| RefinePose | ViT-L | 97.8 | 97.7 | 94.4 | 91.6 | 93.2 | 92.4 | 90.0 | 94.1 |
| RefinePose | ViT-H | 97.9 | 97.8 | 94.6 | 91.3 | 93.4 | 92.7 | 90.3 | 94.2 |

On the whole, the increase of RefinePose is not obvious compared with ViTPose, and the improvements are generally between 0.2 PCKh and 0.4 PCKh. The detection accuracy of RefinePose for "Head" and "Shoulder" is obviously higher than that for other joints, while the detection accuracy of RefinePose for "Wrist" and "Ankle" is the worst. This shows that although the existing human pose estimation methods are already excellent, we are still far from fully detecting all the joints. There is still a lot of work to be done in the research of human pose estimation.

*4.4. Ablation Experiments*

To validate the effectiveness of each of our proposed RefinePose, we conduct the following ablation experiments. We use ViT-B as the backbone of RefinePose in the following experiments. The validations of GradualEmbedding and the Fusion Decoder are performed on the COCO val2017 dataset, and the validation of the shortcut is performed on the MPII dataset. Other experimental configurations remain unchanged. Experiments follow the single-variable principle.

4.4.1. Effectiveness of the GradualEmbedding Module

To verify the effectiveness of our proposed GradualEmbedding module, we conduct ablation experiments on each part of the GradualEmbedding module, and the experimental results are shown in Table 3.

**Table 3.** Ablation experiment of the GradualEmbedding module.

| DownSample with Conv2d | DownSample with MaxPool | Feature Aggregation | AP | AR |
|---|---|---|---|---|
| ✓ | - | - | 73.1 | 79.2 |
| - | ✓ | - | 72.6 | 78.3 |
| ✓ | ✓ | - | 74.5 | 80.2 |
| ✓ | ✓ | ✓ | 75.3 | 80.5 |

As can be seen from Figure 2, GradualEmbedding is mainly composed of four Down-Sample stages, and each stage consists of ScaleBlock and FeatureAggregation. Additionally, the output of ScaleBlock is the sum of the results of the max pooling layer and convolutional layer. From the table Table 3, we can see that when we only use the convolutional layer with stride 2 for downsampling, the accuracy of RefinePose is only 73.1 AP. When we only use the max pooling layer for downsampling, the recognition accuracy further drops to

72.6 AP. The accuracy of the model is not satisfactory if only using convolution or max pooling for downsampling. However, if we integrate the two parts, RefinePose can achieve a recognition accuracy of 74.5 AP. A reasonable explanation is that the max pooling layer can extract features with high response values in the feature map so that the feature map retains more meaningful information in the process of downsampling. The convolution layer extracts local features in the downsampling process. By fusing the results of the max pooling layer and the convolutional layer, we can obtain a more informative feature representation. After adding the FeatureAggregation module, the accuracy of RefinePose is further improved to 75.3 AP, which increased by 0.8 AP. This shows that feature aggregation is meaningful after downsampling. The above ablation experiments demonstrate that our designed GradualEmbedding is effective.

### 4.4.2. Effectiveness of the Fusion Decoder

In order to verify the effectiveness of our proposed Fusion Decoder, we conduct ablation experiments on each part of the Fusion Decoder. The experimental results are shown in Table 4.

**Table 4.** Ablation experiment of the Fusion Decoder. FA denotes the FeatureAggregation module.

| UpSample | AP | AR |
|---|---|---|
| Bilinear | 74.2 | 79.6 |
| Bilinear + MaxUnpool | 74.9 | 80.1 |
| Bilinear + MaxUnpool + FA | 75.3 | 80.5 |

The Fusion Decoder is mainly composed of bilinear interpolation, MaxUnpool layer, and the FeatureAggregation module, as shown in Figures 1 and 4. We explore the impact of the above three parts on the performance of RefinePose, as shown in Table 4. As can be seen from the table, RefinePose obtains 74.2 AP by using bilinear interpolation to upsample. The recognition accuracy of RefinePose reaches 74.9 AP after adding the MaxUnpool layer, which is further improved by 0.7 AP. This shows that the combination of the MaxUnpool layer and bilinear interpolation for upsampling can bring better performance. Adding FeatureAggregation to the Fusion Decoder also improves the accuracy, reaching 75.3 AP. Table 4 demonstrates the value of the Fusion Decoder.

### 4.4.3. Effectiveness Of The Shortcut

The Transformer needs a lot of training data to fully exploit its performance. However, in some situations, there may not be enough training data for us to use. To this end, we add shortcuts in the stacking module of transformers to alleviate the problem of insufficient training on small datasets. We conduct ablation experiments to explore the effectiveness of adding shortcuts, and the experimental results are shown in Table 5. As can be seen from the table, the recognition accuracy of RefinePose is improved from 93.2 PCKh to 93.6 PCKh after adding the shortcut. The improvement of 0.4 PCKh indicates that adding shortcuts is valuable. Although the improvement is weak, it also alleviates the problem of insufficient training of large-scale transformer-based networks on small datasets to some extent.

**Table 5.** Ablation experiment of the shortcut.

| Shortcut | Mean (PCKh@0.5) |
|---|---|
| - | 93.2 |
| ✓ | 93.6 |

### 4.5. Comparison with the ViTPose

In order to compare the excellent performance of RefinePose more intuitively, we test the ViTPose and RefinePose on the validation set of the COCO dataset, respectively, and

randomly select several images from the results for comparison. The test results are shown in Figure 6.

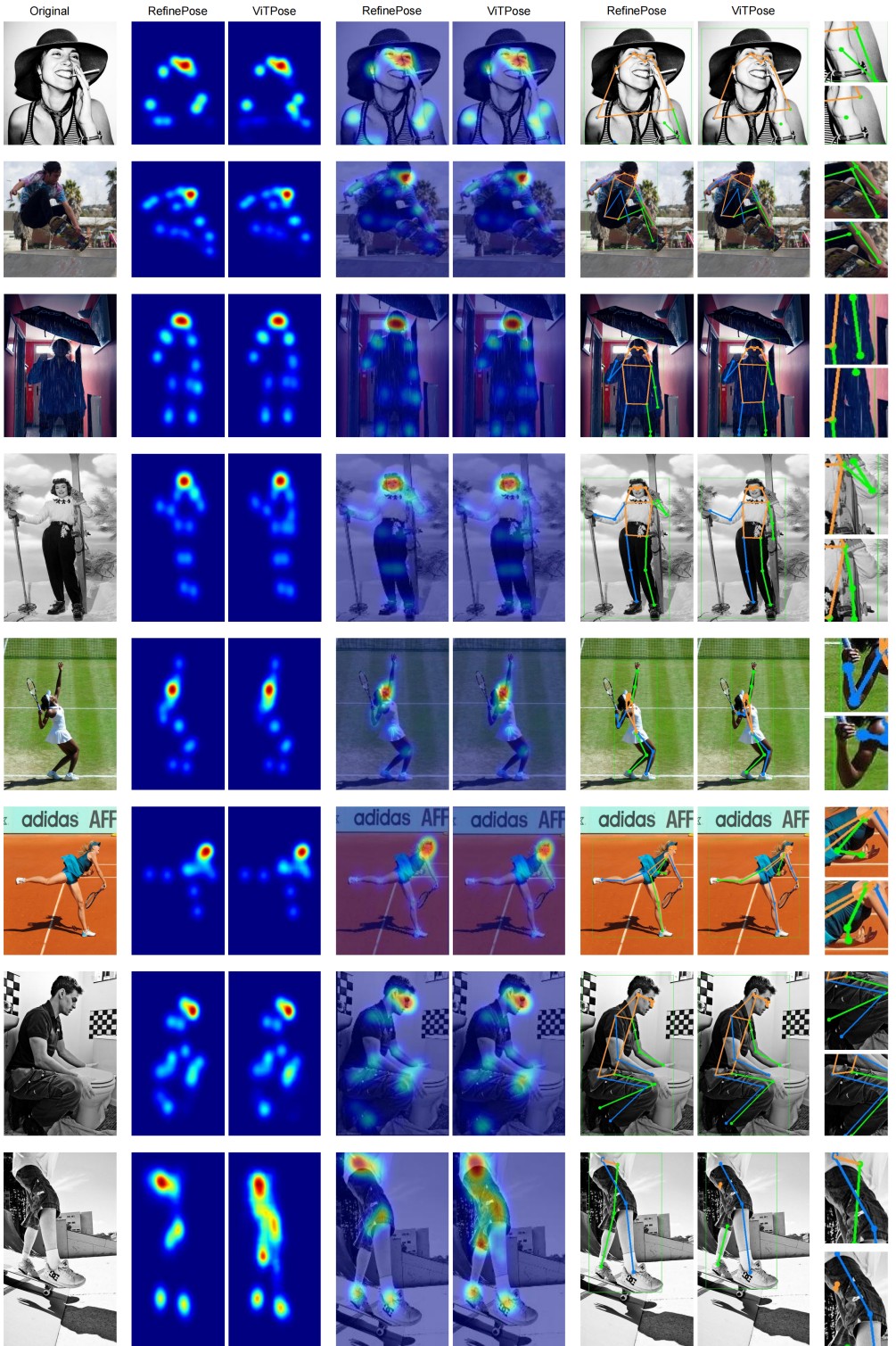

**Figure 6.** Performance comparisons between ViTPose [1] and our RefinePose. There are 5 columns from left to right, which are the original images, the feature visualizations, the fusion results between feature maps and the original images, the keypoint prediction results, and the detail comparisons.

As can be seen from Figure 6, benefits from the outstanding design of GradualEmbedding and Fusion Decoder, RefinePose obtains more refined prediction results than ViTPose.

In the case of occlusion and complex actions, the prediction results of ViTPose are deviated or even not predicted, while RefinePose can predict the joint points of the occluded parts well. In addition, we can see from the second column that the heatmaps of RefinePose (left) are more concise and neat than the ViTPose (right), and the response values of joints are more prominent. Figure 6 demonstrates the superior performance of RefinePose in the case of occlusion and complex actions.

## 5. Conclusions

In this paper, we propose a more refined pose estimation network based on ViTPose, termed RefinePose. First, we design a novel GradualEmbedding to replace the original PatchEmbedding. GradualEmbedding uses a progressive downsampling strategy to gradually reduce the size of feature maps, which alleviates the problem of feature information loss. Second, we propose a more refined decoding method, named Fusion Decoder. Compared with the Simple Decoder and the Classic Decoder in ViTPose, the Fusion Decoder combines max unpooling and bilinear interpolation to accurately restore the predicted heatmap. Third, to alleviate the problem of insufficient training of transformers on small datasets, we add shortcuts to the stacking module of transformers. We validate the performance of RefinePose on the COCO dataset and MPII dataset and obtain better recognition accuracy than ViTPose. We hope that our method can make some contributions to the development of human pose estimation.

**Author Contributions:** Conceptualization, G.W. and H.D.; methodology, C.C.; software, H.D.; validation, H.D.; formal analysis, X.Z.; investigation, H.D. and X.Z.; resources, X.Z.; data curation, X.Z.; writing—original draft preparation, H.D.; writing—review and editing, C.C.; visualization, X.Z.; supervision, G.W. and C.C.; project administration, G.W.; funding acquisition, G.W. All authors have read and agreed to the published version of the manuscript.

**Funding:** This research was funded by Youth Innovation and Technology Support Plan of Colleges and Universities in Shandong Province grant number 2021KJ062.

**Data Availability Statement:** The public datasets used in this study are COCO and MPII. Details about COCO can be found here in this link: https://cocodataset.org (accessed on 11 November 2022). Details about COCO can be found here in this link: http://human-pose.mpi-inf.mpg.de (accessed on 11 November 2022).

**Acknowledgments:** Youth Innovation and Technology Support Plan of Colleges and Universities in Shandong Province (2021KJ062).

**Conflicts of Interest:** The authors declare no conflict of interest. The funders had no role in the design of the study; in the collection, analyses, or interpretation of data; in the writing of the manuscript; or in the decision to publish the results.

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
