# Peer review of "RefinePose: Towards More Refined Human Pose Estimation"

_electronics, doi:10.3390/electronics11234060_

Round 1

Reviewer 1 Report

Please explain the main reason you selected VITPose is?  

The idea to propose GradualEmbeding to replace PathEmbeding is good. 

 Do you have an experiment on how FAST the algorithm is ?

Author Response

1,Thank you very much for your question. We chose ViTPose because it is the latest human posture estimation method proposed in 2022. And on the papers-with-code website (https://paperswithcode.com/sota/pose-estimation-on-coco-test-dev), it is the method with the highest recognition accuracy in human posture estimation task.

2,Thank you very much for your question. We have not evaluated the speed of the algorithm before. But after receiving your suggestion, we test the speed of the model. We test the model of the base version on the COCO val2017 dataset. RefinePose runs at 101 fps. For comparison, the speed of ViTPose under the same configuration is 184 fps. Although the precision of RefinePose is higher than ViTPose, it runs slower. Further optimizing the performance of RefinePose is the direction of our next work. Thank you again for your suggestions.

Reviewer 2 Report

In this paper, the authors describe “RefinePose: Towards More Refined Human Pose Estimation”. It can become an interesting paper for electronics after major revision. Followings are my comments.

(1)  To easily realize the implementation details, authors have to explain how many images are produced after data augmentation methods.

(2)  And, what kind of pose processing is done with UDP? Authors have to explain it in detail.

(3)  In Table 2, the results of RefinePose method are slightly better than ViTPose. However, the results of RefinePose method are significantly better than ViTPose in Figure 6. Authors have to explain this discrepancy in results.

(4)  Authors cite many conference papers and reposts. Reviewer suggests that authors cite some important journal papers of human pose estimation.

(5)  Style of references is not in agreement with this Journal style, and thus it requires a revision.

Author Response

1, Thank you for your suggestion. The COCO dataset has 118287 images in the train2017 set and 5000 images in the val2017 set. Before sending the images into the model, we carry out data augmentation on the images. These augmentation methods include random flipping, random rotation, scaling, regularization, etc. These data augmentation methods only change the size, rotation angle, or mean value of images, and do not change the number of images. Therefore, the number of the training set and Val set after data augmentation is still 118287 and 5000, respectively. We will explain this point in the revised paper.

2, Thank you for your question. UDP is a commonly used data processing method in human pose estimation, so we haven't explained it too much. After receiving your suggestion, we briefly explained the idea of UDP in the revised paper. The main idea of UDP includes two aspects: one is to use the unit length to measure the size of images instead of the number of pixels during data processing. The other is to introduce an encoding and decoding method without statistical error under ideal conditions. If readers want to know more about the UDP method, we recommend readers to read the original paper. After all, this is not the focus of our article.

3, Thank you for your question. We have the following two answers to your question. (1) The improved RefinePose has better recognition performance than ViTPose, which is beyond doubt. However, as shown in Table 2, the recognition accuracy of RefinePose is slightly higher than the ViTPose, increasing by 0.2 points on the MPII dataset (with ViT-B backbone). This small increase may not be obvious testing on a few images. But when there are enough images to be tested, this small increase in accuracy is obviously shown. The COCO dataset has 118287 images in the train2017 set. It is easy to find 9 images whose effect of RefinePose is better than ViTPose. (2) In Figure 6, we have magnified the local area to more clearly compare the performance difference between RefinePose and ViTPose. However, in the original recognition results, the difference between the two methods may be so small that this difference is not reflected in the algorithm evaluation. Overall, the performance of the two methods is excellent, and the recognition results for most images are very good. But for a few images such as complex movements or occlusions, the advantages of RefinePose are reflected.

4, Thank you very much for your advice. We have added some articles from important journals to our paper.

5, Thank you very much for your suggestion. We have modified the reference style in the paper, according to the official latex template provided by Electronics.

Reviewer 3 Report

-----------------------

Summary

-----------------------

The paper « RefinePose: Towards More Refined Human Pose Estimation » proposes a VIT-based model for human pose estimation, called RefinedPose. 

This model relies on the ViTPose and contributes with 3 additions : a novel GradualEmbedding module, a fusion decoder module and shortcuts on the Transformers architecture.

Two experiments show challenging results on COCO and MPII datasets.

-----------------------

Comments

-----------------------

The paper is well written, well organized and easy to understand.

The study context is clearly introduced. 

The related works are adequate.

The proposed method is sufficiently described nevertheless the main inspirations should be more precisely argued.

These two sentences should be discussed :

"Inspired by Odena et al. [6], we design a new decoding method, termed Fusion Decoder."

"inspired by ResNet [23], we divide each four transformer modules into a group and add shortcuts to each group"

About shortcuts, only one way to build shortcuts is presented. 

So no comparative studies are shown. 

The transformer modules could be divided in different fashions, so what could be the performance impacts ?

The experimental part is clear with convincing results and the ablation part is effective. 

Author Response

1, Thank you very much for your question. We explain the relevant inspirations in detail as follows and add them to the revised paper.
“Odena et al. point out in the article [1] that the deconvolution operation makes the generated image have a chessboard effect, and the combination of interpolation and convolution can eliminate this chessboard effect. Therefore, inspired by Odena et al., we design a new decoding method, termed Fusion Decoder.”
“When the depth of the network increases continuously, the accuracy of the model will saturate or even decline. The ResNet model [2] proposes a residual structure, allowing the output of the model to skip some middle layers and directly participate in the training of later layers. This residual method can avoid gradient disappearance and speed up network training. Inspired by ResNet [2], we divide each four transformer modules into a group and add shortcuts to each group.”

2, Thank you for your question. At the early stage of designing RefinePose, we tried several different divided fashions of transformers (i.e., different connection methods of shortcuts. Because the division of transformers depends on the connection method of shortcuts.), such as the interval skip connection of ResNet [2], the dense connection of DenseNet [3], and the hourglass connection of Hourglass [4]. However, the experimental results of these divided fashions were not ideal, so we abandoned them. After a lot of experiments, the divided fashion of transformers proposed in the current paper was selected. Unfortunately, in the early experiments, we did not record the experimental data at that time, so we can not give a rigorous data comparison here. But the question you raised is very research worthy. In the future, we will continue to explore more effective divided ways.

Round 2

Reviewer 2 Report

Reviewer recommends to accept without comments.